# Geographic variation in cardiometabolic risk distribution: A cross-sectional study of 256,525 adult residents in the Illawarra-Shoalhaven region of the NSW, Australia

**Renin Toms**[1,2]*, **Darren J. Mayne**[1,2,3,4], **Xiaoqi Feng**[2,5,6], **Andrew Bonney**[1,2]

**1** School of Medicine, University of Wollongong, Wollongong, NSW, Australia, **2** Illawarra Health and Medical Research Institute, Wollongong, NSW, Australia, **3** Public Health Unit, Illawarra Shoalhaven Local Health District, Warrawong, NSW, Australia, **4** School of Public Health, The University of Sydney, Sydney, NSW, Australia, **5** School of Public Health and Community Medicine, University of New South Wales, Sydney, NSW, Australia, **6** Population Wellbeing and Environment Research Lab (PowerLab), School of Health and Society, University of Wollongong, Wollongong, NSW, Australia

* rmbst288@uowmail.edu.au

**Data Availability Statement:** Access to, and use of, Southern IML Research (SIMLR) Study data are subject to a License Agreement — Provision of

## Abstract

### Introduction

Metabolic risk factors for cardiovascular disease (CVD) warrant significant public health concern globally. This study aims to utilise the regional database of a major laboratory network to describe the geographic distribution pattern of eight different cardiometabolic risk factors (CMRFs), which in turn can potentially generate hypotheses for future research into locality specific preventive approaches.

### Method

A cross-sectional design utilising de-identified laboratory data on eight CMRFs including fasting blood sugar level (FBSL); glycated haemoglobin (HbA1c); total cholesterol (TC); high density lipoprotein (HDL); albumin creatinine ratio (ACR); estimated glomerular filtration rate (eGFR); body mass index (BMI); and diabetes mellitus (DM) status was used to undertake descriptive and spatial analyses. CMRF test results were dichotomised into 'higher risk' and 'lower risk' values based on existing risk definitions. Australian Census Statistical Area Level 1 (SA1) were used as the geographic units of analysis, and an Empirical Bayes (EB) approach was used to smooth rates at SA1 level. Choropleth maps demonstrating the distribution of CMRFs rates at SA1 level were produced. Spatial clustering of CMRFs was assessed using Global Moran's I test and Local Indicators of Spatial Autocorrelation (LISA).

### Results

A total of 1,132,016 test data derived from 256,525 individuals revealed significant geographic variation in the distribution of 'higher risk' CMRF findings. The populated eastern seaboard of the study region demonstrated the highest rates of CMRFs. Global Moran's I

Data (LA) between Southern IML Pathology Pty Ltd (Data Owner) and The University of Wollongong (License Holder), and a Data Access Agreement (DAA) between the License Holder and researchers (Data Users). This process is facilitated by the Illawarra Health and Medical Research Institute (IHMRI) (Data Custodian) through the Southern IML Research Study — Cohort Management Committee (SIMLR—CMC). The Data License does not allow for "public access" to data; however, researcher may access to SIMLR Study data subject to approval by the SIMLR—CMC and an appropriately constituted Australian Human Research Ethics Committee (HREC) as defined in the National Health and Medical Research Council's National Statement on Ethical Conduct in Human Research (2007) (available from https://www.nhmrc.gov.au/about-us/publications/national-statement-ethical-conduct-human-research-2007-updated-2018). The Data License requires at least one of the research team be affiliated with IHMRI. SIMLR—CMC contact details are: C/o- Associate Professor Kathryn Weston; Southern IML Research Study — Cohort Management Committee; Illawarra Health and Medical Research Institute; Building 32, University of Wollongong, Northfields Avenue, Wollongong NSW 2522, Australia; Phone +61 2 4221 4333; Email: info@ihmri.org.au; Web Link: https://www.ihmri.org.au/research-projects/simlr-cohort-study/.

**Funding:** The Australian Government Research Training Program Scholarship covers the university tuition fees of the PhD candidate who is the principal author in this study (RT).

**Competing interests:** The authors have declared that no competing interests exist.

values were significant and positive at SA1 level for all CMRFs. The highest spatial autocorrelation strength was found among obesity rates (0.328), and the lowest for albuminuria (0.028). LISA tests identified significant High-High (HH) and Low-Low (LL) spatial clusters of CMRFs, with LL predominantly in the less populated northern, central and southern regions of the study area.

## Conclusion

The study describes a range of CMRFs with different distributions in the study region. The results allow generation of hypotheses to test in future research concerning location specific population health approaches.

## Introduction

Uncontrolled cardiometabolic risk factors (CMRFs) such as hyperglycaemia, dyslipidaemia, albuminuria, inadequate glomerular filtration, overweight and/or obesity and diabetes can predispose and heighten the risk for cardiovascular disease (CVD).[1–6] Cardiovascular diseases are the leading cause of death worldwide, and the highest absorber of health care expenditure in many developed nations, including Australia.[7–9]

In Australia, CVD remain the single leading cause of death; the largest health problem; and a major economic burden.[10,11] Nine in 10 adult Australians have at least one CVD risk factor, and one in four have three or more risk factors.[11] CVD kills one Australian every 12 minutes and one in six Australians (3.7 million people) are thought to be at risk.[12] In addition, the prevalence of CVD is projected to steeply increase in the coming decades.[11] A deceleration in the rapid growth of this major health care issue is possible only through the prevention and control of CMRFs. The role of CMRFs in the population, over and above individual level factors such as age, are being questioned in regard to discriminatory accuracy for development of CVD.[13] However identification of one or more CMRFs in a person at any age can initiate preventive lifestyle changes which may have significant benefits.[14–18] Similarly, identification of areas with higher rates of CMRFs can potentially trigger further area-level analyses investigating the potential for targeted health service commissioning.[19–21]

Advances in Geographic Information System (GIS) over the last quarter of a century have provided various tools to integrate epidemiological and geographical data.[22–24] Geocoding of risk parameters became feasible with such tools for its area-level analyses, which has facilitated area-level mapping of risk parameters, which has the potential generate hypothesis for regional health care research.[22] Thus integrating risk parameters through GIS has the potential to facilitate area-level health research, [25–28]; however, not without potential pitfalls [29–31]. A limitation of GIS-based mapping is that its outputs may be misleading, especially if maps are not smoothed using appropriate spatial or multilevel analyses.[32–34] However, it is well recognised in the literature that area level community interventions based on GIS approaches have been successful in a number of countries. [19–21,35,36]

There has been a significant increase in the number of epidemiological studies using spatial analytical methods in the last decade, including international studies reporting significant geographic variation in CMRFs at different spatial scales of measurements.[37–45] Hyperglycaemia was the most commonly reported CMRF displaying variation, followed by dyslipidaemia, overweight and/or obesity, and inadequate glomerular filtration.[37] Multiple risk factors were rarely analysed in these studies, though most CMRFs are interrelated and often coexist.[46] In

this study, we aim to demonstrate the feasibility of utilising laboratory based routine test data to generate basic distribution maps of eight different CMRFs in regional New South Wales (NSW), Australia. The research questions we address are: (1) what is the geographic distribution pattern of CMRFs in the study area; and (2) is there any significant spatial clustering of CMRFs rates? The research sought to identify area-level patterns in the distribution of CMRFs that could be used to generate hypotheses for future research with the goal of improving health service commissioning in the study region.

## Methods

The study adopted a cross-sectional design and was approved by the University of Wollongong (UOW) and Illawarra and Shoalhaven Local Health District (ISLHD) Human Research Ethics Committee (HREC 2017/124).

### Setting

The study was undertaken in the Illawarra-Shoalhaven region (ISR) of the NSW, Australia. The ISR region stretches from the immediate south of the metropolitan boarders of Sydney, and extends along the south-eastern coastal belt of NSW—bordered by the Pacific Ocean in the east and the coastal escarpment of the Southern Tablelands in the West. This region encompasses multiple cities, towns and rural areas and includes the four local government areas of Wollongong, Shellharbour, Kiama and Shoalhaven. Overall, the ISR covers a land area of 5615 square kilometres and had an estimated residential population of 369,469 persons at the 2011 Australian Census of Population and Housing, of which 285, 385 (77.24%) were adults ($>$ = 18 years).[47] De-identified data for this study were obtained from the Southern IML Research (SIMLR) Study, a large-scale community-derived cohort of internally-linked and geographically referenced pathology data collected in routine practice by the largest pathology provider servicing the study area. More details on this data source, its access and maintenance are published elsewhere.[48]

Statistical Area level 1 (SA1) was used as the geographic unit of analysis in this study, which was the smallest geographic unit for the release of Census data in 2011.[49] SA1s generally have a population of 200 to 800 persons (400 average), and the ISR includes a total of 980 conterminous SA1s. Fig 1 shows the study area with SA1 units and the major landmarks of the region. Very small and crowded SA1s similar to the areas shown the inset map tend to be more densely populated.

### Participants and variables

The CMRF test data of the adult residents of ISR between 1 Jan 2012–31 Dec 2017 (6 years) were extracted for analyses from the SIMLR database. Test data were extracted for eight CMRFs: fasting blood sugar level (FBSL); glycated haemoglobin (HbA1c); total cholesterol (TC); high density lipoprotein (HDL); albumin creatinine ratio (ACR); estimated glomerular filtration rate (eGFR); body mass index (BMI) and diabetes mellitus (DM) status. The SIMLR database uses an algorithm to identify DM status based on diagnosis guidelines published by the Royal Australian College of General Practitioners (RACGP) and Diabetes Australia, and methods from the National Health Survey of the Australian Bureau of Statistics (ABS).[50,51] The algorithm identifies DM for HbA1c $\geq$ 6.5% or FBSL $\geq$ 7.0 mmol/l within +/- 24 months of HbA1c $<$ 6.5%. The study data included both prevalent and incident DM cases. Study data included only the most recent CMRF test result for each individual. We excluded extreme BMI values $<$12 and $>$80 based on cut-off points reported by Cheng (2016), Li (2009) and Littman (2012).[52–54] **Table 1** lists the CMRFs value definitions adopted in this study and their source references.

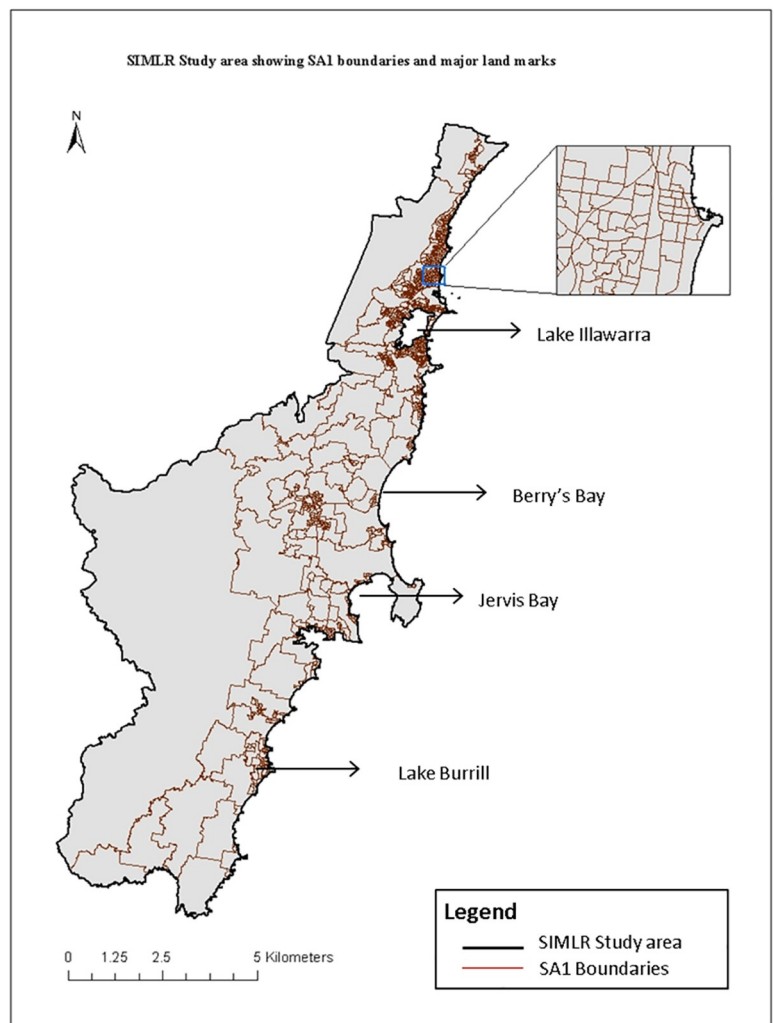

**Fig 1. Map of the Illawarra-Shoalhaven region of NSW Australia showing SA1 areas and major landmarks.**

## Statistical and spatial analyses

First, individual-level descriptive analyses of CMRFs were performed. The total number of each CMRF tests and summary statistics of each tests' results are reported. The summary

**Table 1. Cardiometabolic risk classification.**

| 'Higher risk' CMRFS | Value definition | Adopted from |
|---|---|---|
| High FBSL | FBSL ≥7.0 mmol/l | RACGP guidelines[50] |
| High HbA1c | HbA1c > 7.5% | RACGP guidelines[50] |
| High TC | TC ≥ 5.5 mmol/l | Australian Health Survey[55] |
| Low HDL | HDL < 1 mmol/l[56] | National heart foundation of Australia[57] |
| High ACR | ACR ≥ 30 mcg/L to mg/l | Kidney Health Australia[58] |
| Low eGFR | eGFR < 60 mL/min/1.73m2 | Kidney Health Australia[58] |
| High BMI | BMI ≥ 30 (Obese) | World Health Organization (WHO)[59] |
| DM Status | +ve DM test algorithm | RACGP guidelines[50] and Australian Health Survey[55] |

values for eGFR test results are calculated using the approach for grouped data as eGFR test result values are truncated at >90 in the SIMLR Study data. Test results were dichotomised into 'higher risk' and 'lower risk' categories based on the CMRF definitions in Table 1.

Second, area-level analyses of CMRFs were undertaken. Within-cohort prevalence of 'higher risk' CMRF findings are calculated using the total number of tests within each SA1 as the denominator. The exception were DM cases, which are likely to include most prevalent cases in the study area, so SA1 adult populations aged 18 years and over were used as the denominators (accessed from ABS census 2011 data). Thereafter, an Empirical Bayes (EB) approach was used to smooth all the CMRFs' raw rates to minimise extreme values arising from small sample sizes. The EB smoothed rates were then imported into GIS software for mapping and spatial statistical analyses.

As individuals with CMRFs are assumed randomly distributed within the study area, the geographic distribution of CMRFs is assumed spatially independent in this study. Global Moran's I test was used to identify spatial autocorrelation of CMRFs at a 0.05 level of significance. Global Moran's I tests if the geographic distribution of rates is clustered, dispersed or random based.[60] The global Moran's I also indicates the general strength of spatial autocorrelation in the study area, which theoretically ranges between -1 to +1. Values of I significantly above -1/(N-1) indicate positive spatial autocorrelation, where N is the number of spatial units indexed. [61] When significant spatial autocorrelation was detected, Local Indicator of Spatial Autocorrelation (LISA) spatial statistics were used to identify any clustering of CMRFs.[62] LISA was used to indicate spatial clustering of High-High (HH) or Low-Low (LL) CMRFs rates at SA1-level within the study region. False Discovery Rate (FDR) corrections were applied to LISA tests to correct p-values for multiple testing.

All descriptive statistics and EB smoothing were performed using R version 3.4.4.(R Foundation for Statistical Computing, Vienna, Austria).[63] Mapping and spatial analyses were performed using ArcGIS version 10.4.1(ESRI Inc. Redlands, CA, USA).[64]

## Results

The study sample comprised 1,132,016 test results contributed by 256,525 adult individuals residing in the study region. Of the 256,525 individuals, 193,679 (75.5%) had FBSL, 73,885 (28.8%) had HbA1, 194,816 (75.9%) had TC, 182,237 had HDL (71.0%), 50,790 had ACR (19.8%), 244,166 had eGFR (95.2%), and 192,443 had BMI (75.0%) test results. It was estimated 23,704 (9.2%) of persons met the clinical criteria for diabetes. Table 2 provides the summary statistics of CMRF test results.

The CMRF test result values were dichotomised into 'higher risk' and 'lower risk' categories based on the CMRF definitions in Table 1. The proportion of individuals with 'higher risk' CMRFs findings varied considerably between tests. The largest 'higher risk' proportions were

**Table 2. Summary statistics of CMRFs test results.**

| CMRFs | Tests | Mean | SD | Min | 1st Qu | Median | 3rd Qu | Max |
|---|---|---|---|---|---|---|---|---|
| FBSL | 193679 | 5.6 | 1.6 | 0.7 | 4.9 | 5.3 | 5.8 | 43.9 |
| HbA1c | 73885 | 6.0 | 1.3 | 2.6 | 5.3 | 5.6 | 6.4 | 17.8 |
| TC | 194816 | 5.0 | 1.1 | 1.1 | 4.2 | 4.9 | 5.7 | 39.4 |
| HDL | 182237 | 1.5 | 1.2 | 0.1 | 0.5 | 1.4 | 1.8 | 5.8 |
| ACR | 50790 | 7.4 | 40.3 | 0.1 | 0.4 | 0.8 | 2.3 | 1291.5 |
| eGFR | 244166 | 75.8 | 13.8 | 2.0 | - | 83.2 | - | >90.0 |
| BMI | 192443 | 28.4 | 6.1 | 12.0 | 24.1 | 27.5 | 31.6 | 78.1 |

found for BMI (33.74%) and TC (32.55%), and the lowest for ACR (4.03%). Table 3 provides details on the CMRF test results classification and the identified proportions.

## Geographic distribution of cardiometabolic risk factors

Fig 2 shows the geographic distribution of CMRFs at SA1 level in the ISR region with red indicating the highest and blue the lowest rates of risk. SA1s with no test data appear in white. Areas with higher rates of CMRFs were found to be clustering within the study region. The highest rates were found mainly along the populated eastern board of the study region; notably among SA1s around Lake Illawarra, south-east of Berry's bay, and east of Lake Burill. However, the high TC rates showed a reversed pattern, and higher rates were found in the relatively less populated central and westerly aspects of the study area. HDL rates did not follow this reversed pattern.

## Spatial autocorrelation of CMRFs

The global Moran's I tests were significant and positive for all CMRFs (Table 4). The highest spatial autocorrelation strength was found among obesity rates (0.328), followed by high FBSL (0.184) and low HDL (0.174). The spatial autocorrelation strength was the lowest for albuminuria (0.028) and low eGFR (0.069).

LISA tests identified significant spatial clustering of CMRFs in the ISR region. The HH clusters were found mainly along the populated areas of the study region, except for TC. Areas around the immediate surroundings of Lake Illawarra had the most HH clusters, followed by the areas to the south-west of Berry's Bay and south of Jervis Bay. A few areas around Lake

**Table 3. Frequency and proportion of 'higher risk' results of CMRFs tests.**

| Cardiometabolic risk | Classification | Tests n (%)* |
|---|---|---|
| FBSL | | 193679 (100) |
| FBSL ≥7.0 mmol/L | Higher risk | 16280(8.4) |
| FBG < 7.0 mmol/L | Lower risk | 177399(91.6) |
| HbA1c | | 73885(100) |
| HbA1c > 7.5% | Higher risk | 7927(10.7) |
| HbA1c ≤ 7.5% | Lower risk | 65958(89.3) |
| TC | | 194816(100) |
| TC ≥ 5.5 mmol/L | Higher risk | 63422(32.5) |
| TC < 5.5 mmol/L | Lower risk | 131394(67.5) |
| HDL | | 182237 (100) |
| HDL < 1 mmol/l | Higher risk | 21261(11.7) |
| HDL ≥ 1 mmol/l | Lower risk | 160976(88.3) |
| ACR | | 50790(100) |
| ACR ≥30 mcg/L to mg/L | Higher risk | 2047 (4.1) |
| ACR <30 mcg/L to mg/L | Lower risk | 48743(95.9) |
| eGFR | | 244166(100) |
| eGFR < 60 mL/min/1.73m$^2$ | Higher risk | 27241(11.2) |
| eGFR20 ≥ 60 mL/min/1.73m$^2$ | Lower risk | 216925(88.8) |
| BMI | | 192455(100) |
| BMI ≥ 30 (Obesity) | Higher risk | 64832(33.7) |
| BMI < 30 | Lower risk | 127511 (66.3) |

* The denominators for percentages are the total number of each CMRFs tests

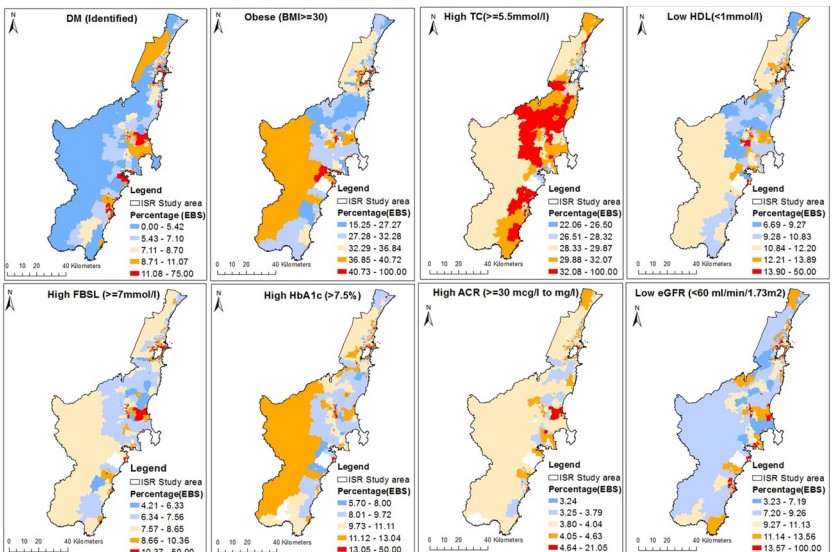

**Fig 2. Geographic distribution of the proportion of CMRFs within the Illawarra Shoalhaven region of the NSW Australia.**

Burrill had HH clusters of DM, TC and eGFR. The LL clusters were mainly around the less populated north, central and south ends of the study area, except for TC. The TC clusters demonstrated a reverse pattern in comparison with all other CMRFs, where HH clusters were mainly around the less populated central and southern ends of the ISR and a few instances in the north-eastern end of the study area. LL clusters of TC were found around the immediate surroundings of Lake Illawarra. Fig 3 illustrates the spatial clustering of CMRFs in the study area.

## Discussion

Place has always been a key element in human health and epidemiology. In the present study, we explored the geographic distribution of eight CMRFs in 980 SA1s in a regional area of NSW, Australia. The study is a first of its kind known to us in providing a comprehensive small area-level profile of a wide range of cardiometabolic risk factors, and provides an example of using population-derived routine laboratory data for area-level research.

Higher rates and clustering of CMRFs were mostly observed along the more densely populated eastern coast line of the study region. Also, some areas were common for multiple risk factors as their distribution pattern frequently converged in these areas, for example areas

**Table 4. Spatial autocorrelation (Moran's I) of CMRFs.**

| CMRFs | Moran's I | z-score | p-value |
|---|---|---|---|
| DM | 0.097 | 27.952 | <0.0001 |
| Obesity | 0.328 | 92.086 | <0.0001 |
| High FBSL | 0.184 | 51.539 | <0.0001 |
| High HbA1c | 0.101 | 28.030 | <0.0001 |
| High TC | 0.146 | 41.154 | <0.0001 |
| Low HDL | 0.174 | 48.733 | <0.0001 |
| Albuminuria | 0.028 | 8.096 | <0.0001 |
| Low eGFR | 0.069 | 19.699 | <0.0001 |

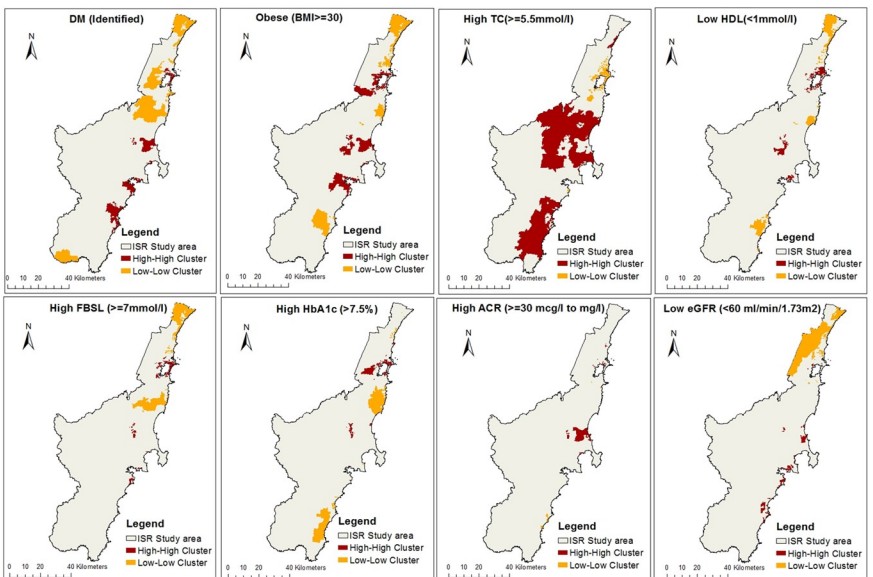

**Fig 3. Local Moran's I cluster maps showing high-high and low-low spatial associations of CMRFs within the Illawarra Shoalhaven region of the NSW Australia.**

around Lake Illawarra and south of Jervis Bay. However, not all populated areas were involved in this pattern, and some less populated areas also had higher rates of risk. Spatial analyses revealed significant spatial autocorrelation for all eight CMRFs. Patterns of clustering were different for each CMRFs at the small-area scale used in this study, which provides directions for future research using multilevel analytic methods.[65]

The distribution of high TC values were generally reversed to those distributions of other CMRFs described in this study. The reason for this observation is yet to be explored, but a possible treatment effect is suspected as the lower risk areas were often densely populated areas. It is possible that the people residing in these areas have better access to health care services and more frequently prescribed anti-cholesterol drugs.[66,67] However, not all densely populated areas were involved in this 'higher risk' TC distribution pattern and further research is required.

The current study adds to the limited studies from Oceania reporting on geographic variation of CMRFs, and the first from regional Australia. Previous studies from Australia have reported geographic variation of 42% in the odds of being diagnosed with DM among adults living in Sydney.[38] Another study reported geographic variation in glycated haemoglobin (HbA1c) values across 767 Census Collection Districts (CDs) in Adelaide. [44] The study builds on previous research by investigating the distribution of a wide range of CMRFs, which appears to be unique in the literature.

This study must be considered within its limitations. First, the cross-sectional design of the study precludes causal inference. Second, the descriptive analyses performed in this study indicate only significant variations in the geographic distribution of CMRFs, but does not differentiate the individual and/or area-level attributes which might be contributing to this variation. [13] Third, the maps include areas with no test data. Fourth, the study data were obtained from people attending health care services; therefore its point-estimates may not be representative of the general population. Fifth, we cannot exclude the possibility that a higher proportion of positive tests in an area could be due to greater access to pathology services; however exploring this possibility was beyond the scope of the current study.

Future research is required to understand the reasons for the geographic variation reported in this paper. The findings reported in this study suggest hypotheses that will be further explored using appropriate multilevel/hierarchical analyses to differentiate and quantify the individual and area-level contributions to this variation.[65,68–70] Such hierarchical analyses will have the potential to inform development of appropriate area-level health care service policy initiatives. It is important to differentiate the contributions of individual (e.g. age, sex, etc.) and area (e.g. socioeconomic disadvantage, access or proximity to health care services, etc) level attributes to the different patterns of clustering to inform targeted area-level preventive interventions and future health service commissioning decisions to these areas.

In conclusion, area-level descriptive analyses of CMRFs have the potential to highlight inequalities in the geographic distribution of CMRFs. Regional planning for the prevention and management of CMRFs requires information about its epidemiology within specific communities or areas. Centralised approaches of disease prevention and management may not suit regional requirements as the disease pattern in regional areas may differ to those in metropolitan areas and cities. Area specific evidence through regional health care research is important to inform health care service commissioning for area specific decisions and policy developments. This paper demonstrates an initial step in such regional health care research, and a feasible method using population data derived from routine clinical practice.

## Acknowledgments

We would like to thank Southern IML Pathology and staff for their generosity in providing data for the SIMLR Cohort Study and ongoing support. In particular we would like to thank Mr Bryan Jones for providing the technical expertise for the data acquisition and helpful comments on the manuscript. Southern IML Pathology are the owners of the data contained within this publication and the Illawarra Health and Medical Research Institute (IHMRI) is the custodian facilitating access to the data. https://www.ihmri.org.au/research-projects/simlr-cohort-study/

## Author Contributions

**Conceptualization:** Renin Toms, Darren J. Mayne, Xiaoqi Feng, Andrew Bonney.

**Data curation:** Renin Toms, Darren J. Mayne.

**Formal analysis:** Renin Toms.

**Investigation:** Renin Toms.

**Methodology:** Renin Toms, Darren J. Mayne, Andrew Bonney.

**Project administration:** Andrew Bonney.

**Resources:** Andrew Bonney.

**Software:** Renin Toms, Darren J. Mayne.

**Supervision:** Darren J. Mayne, Xiaoqi Feng, Andrew Bonney.

**Validation:** Andrew Bonney.

**Writing – original draft:** Renin Toms.

**Writing – review & editing:** Darren J. Mayne, Xiaoqi Feng, Andrew Bonney.

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
