## [Decision Letter · Decision Letter 0]

9 Jul 2019

PONE-D-19-14817

Geographic variance in cardiometabolic risk distribution: a cross sectional study of 256,525 adult residents in the Illawarra-Shoalhaven region of the NSW, Australia.

PLOS ONE

Dear Toms,

Thank you for submitting your manuscript to PLOS ONE. After careful consideration, we feel that it has merit but does not fully meet PLOS ONE’s publication criteria as it currently stands. Therefore, we invite you to submit a revised version of the manuscript that addresses the points raised during the review process.

We would appreciate receiving your revised manuscript by Aug 23 2019 11:59PM. To enhance the reproducibility of your results, we recommend that if applicable you deposit your laboratory protocols in protocols.io, where a protocol can be assigned its own identifier (DOI) such that it can be cited independently in the future. For instructions see: http://journals.plos.org/plosone/s/submission-guidelines#loc-laboratory-protocols

We look forward to receiving your revised manuscript.

Kind regards,

Wenhao Yu, Ph.D.

Academic Editor

PLOS ONE

Journal Requirements:

This study has been conducted with the support of the Australian Government Research Training Program Scholarship.

Additional Editor Comments:

Dear authors, After taking a look at both the reviewers' comments and your manuscript, I suggest that you carefully respond to the reviewers' comments.

Reviewers' comments:

Reviewer's Responses to Questions

**Comments to the Author**

1. Is the manuscript technically sound, and do the data support the conclusions?

Reviewer #1: No

Reviewer #2: Yes

Reviewer #3: No

2. Has the statistical analysis been performed appropriately and rigorously? 

Reviewer #1: No

Reviewer #2: Yes

Reviewer #3: No

3. Have the authors made all data underlying the findings in their manuscript fully available?

Reviewer #1: No

Reviewer #2: No

Reviewer #3: No

4. Is the manuscript presented in an intelligible fashion and written in standard English?

Reviewer #1: Yes

Reviewer #2: Yes

Reviewer #3: No

5. Review Comments to the Author

Reviewer #1: The authors provided a traditional argumentation on the relevance of CMRFs in CVD. However, the role of traditional risk factors over and above age in the population is being questioned because their low discriminatory accuracy. The low discriminatory accuracy of traditional risk factors is not a mayor problem as far as the treatment of risk factors does not harm. For instance, physical activity, healthy diet, reducing stress and an equitable distribution of resources is always recommended but pharmacological treatment could be questioned. I suggest the authors include a critical perspective in the introduction, rather than the usual “mantra” on the role CMRFs in CVD in the population. Se for instance:

# Merlo J, Mulinari S, Wemrell M, Subramanian SV, Hedblad B. The tyranny of the averages and the indiscriminate use of risk factors in public health: The case of coronary heart disease. SSM Popul Health. 2017;3:684-98.

The above publication refers to many others that may help to understand the idea.

The authors also indicate, “individual level approaches in the past have demonstrated limited success evidenced by its still increasing rates (of CVD), (7–9) area-level community health approaches, in addition, are important”. This is not new, and community level intervention has been launched in many countries since many year. The authors should review this issue and provide some references (to reviews of the literature). However, if you like to quantify the relative “importance” of the areas versus individuals for intervention you should identify the geographical component of the total individual risk variance by applying a multilevel approach. Se for instance

# Merlo J, Asplund K, Lynch J, Rastam L, Dobson A. Population effects on individual systolic blood pressure: a multilevel analysis of the World Health Organization MONICA Project. Am J Epidemiol. 2004;159(12):1168-79.

In addition, a main argument of the authors is that visualizing the geographical distribution of CMRFs has the potential to become a powerful toll for policy makers and health care service planners when planning area-level health care service. Again, this is not new. The use of GIS in health care has a long and well-established tradition. The news could be that the authors have developed an innovative GIS tool in their Region and this is always worthy. I do congratulate the authors. However, the authors should situate their work within this long and well-established GIS tradition.

May central concern, however, is the suitability of the analytical approach in this study. I agree in that “Discerning the distribution pattern of CMRFs in a given area is fundamental to such areal-level targeted approaches”. However, I think this “discerning” is not appropriately done by the use of GIS and spatial analyses. Those approaches are exclusively based on the analysis of area variance but question is when we can consider the area variance as large or important. Statistical significance is not a good criterion. To answer the question you need to discern the share of the total individual variance that is at the area level. For this purpose, you need to perform multilevel models and calculate some measures like the variance partition coefficient (VPC), the area under the curve (AUC) for random effect and even a measure of heterogeneity like the median odds ratio (MOR). Spatial analyses are much better analytical approach than traditional ecological analysis but they are not free of the ecological problems (fallacies, MAUP) and as the ecological analyses, they focus on geographical variation only. The “clustering” in spatial analyses refers to areas not to individuals within areas.

GIS and spatial analysis are fancy and popular because coloured choropleth maps are attractive and easy to understand, and–for decisions makers– the cryptic language and scientific halo of the spatial analyses strength the authority of the information. However, they are ecological analyses with many pitfalls as compared with multilevel analyses. The problem with spatial analysis and GIS is that they may provide misguiding information to decision makers …. That is, just the opposite of what the authors aimed to do…. I refer to other publication for and extended explanation of my critics

# Merlo J, Wagner P, Leckie G. A simple multilevel approach for analysing geographical inequalities in public health reports: The case of municipality differences in obesity. Health Place. 2019;58:102145.

# Merlo J, Viciana-Fernandez FJ, Ramiro-Farinas D, Research Group of Longitudinal Database of Andalusian P. Bringing the individual back to small-area variation studies: a multilevel analysis of all-cause mortality in Andalusia, Spain. Soc Sci Med. 2012;75(8):1477-87.

The authors have a database that seems suitable for multilevel analyses, so my advice is reanalyse the database with this methodology. You can see examples on how to perform the multilevel analyses together with codes and spreadsheets for calculations in previous publications, for instance

# Merlo J, Wagner P, Ghith N, Leckie G. An Original Stepwise Multilevel Logistic Regression Analysis of Discriminatory Accuracy: The Case of Neighbourhoods and Health. PLoS One. 2016;11(4):e0153778.

I also recommended the (free) information provided at the Centre for Multilevel Modelling, Bristol University http://www.bristol.ac.uk/cmm/

You can use the shrunken residuals from the multilevel analysis to create maps but you need to interpret the discriminatory accuracy of those maps by using the VPC and the AUC for instance.

In summary, the authors have had a very worthy initiative by creating a record linkage database in their Region, The GIS and spatial analyses seems well performed but my concern is very fundamental. That is, the spatial GIS analyses are not suitable for your research question and, therefore, the conclusions you provide have not support.

Your results and interpretation may even give misleading information to decision makers, as we do not known the discriminatory accuracy of the maps. To consider targeted area-level preventive interventions and regional health care service planning you need to perform a multilevel analysis as suggested above and considered the idea of proportionate universalism.

Your database seems suitable and I recommend you read the analytical framework proposed above and reanalyse the data accordingly.

Reviewer #2: This manuscript aims to assess spatial clustering patterns of CMRFs in the Illawarra Shoalhaven region of NSW Australia. The methods selected by the authors to do so are well-established ones and the resulting distribution patterns are insightful towards targeted local interventions.

The following may need to be clarified before the manuscript potentially moves forward to publication:

Data source: While the authors note the institutions which own the data & manage access, the link provided seems to be broken (https://www.ihmri.org.au/research-projects/simlr-cohortstudy/) and thus one would be unable to verify pertinent information

Table 2: Summary statistics of ACR & eGFR may need to be re-evaluated; specifically SD value of ACR and 3rd Qu & Max values of eGFR.

Reviewer #3: This paper uses pathology data to present the geographic distributions of a number of cardiovascular disease risk factors. There are problems with the underlying analysis and description of the analysis. These problems are severe enough that they must be addressed prior to a more detailed review.

The denominator used is the paper is unclear and is described differently at multiple points in the paper. The title indicates that this analysis uses 256,385 adult individuals. This value again appears in the results (line 193). However, Table 2 and the methods (line 167) indicate that there are 256,385 tests, not individuals. This is almost certainly the case. The authors report a population of 285,385 adults. The data source is a private pathology service, not a population-level registry or health record. These numbers would then indicate that almost 90% of individuals living in the study area had at least one lab test from this pathology company. Another example of this confusion between number of tests and number of adults is the GFR test. 244,166 tests are reported from 256,525 adults. These values indicate that 85% of the adult population received a GFR test, which almost certainly cannot be correct.

However, if the denominator of the described proportions is the number of tests, then these values cannot be interpreted as the authors have. The authors refer to the calculated measures as “risk”. In epidemiology, this term has a specific meaning that implies using person-time as a denominator. Had the authors used population as a denominator, they would have been calculating prevalence, not risk. However, if (as seems to be true) the authors used the number of tests as the denominator, then these values are simply the proportion of tests meeting a specific criteria. Since these tests were performed on an unhealthy population (as evidenced by 95% having had a kidney function test), these results cannot be generalized to the total population. The results should then be interpreted in light of this.

Overall, the writing in this article is very poor. There are multiple instances of sentences and phrases that are convoluted and have an unclear meaning. I would recommend that an editor and an experienced epidemiologist review this manuscript before publication to clarify the manuscript and to ensure that correct epidemiologic terms are used. One example of this is the last sentence of the introduction. I’m unsure what the authors are trying to convey in that sentence and why that sentence is in the paper. There are multiple, similar examples throughout the paper (e.g. “frequency and proportion of risks” (line 166), “discrete test proportions” (line 194), etc.).

Finally, I question if it is reasonable to assume that individuals in the lab data are equally distributed throughout the study area. Spatial clustering of some results could simply be the result of some areas being over-represented in the data. That is, some locations with a high proportion of positive tests could reflect greater access to the pathology lab in that location.

6. PLOS authors have the option to publish the peer review history of their article (what does this mean?). If published, this will include your full peer review and any attached files.

Reviewer #1: Yes: Juan Merlo

Reviewer #2: No

Reviewer #3: No

---

## [Author Response · Author response to Decision Letter 0]

28 Aug 2019

Response to reviewers’ comments

We sincerely thank our reviewers for the constructive comments and valuable criticisms, which were of great help in revising the manuscript. Please find following our detailed responses (AC) to the reviewers’ comments (RC) below.

1. Response to reviewer # 1 comments:

RC 1.1 The authors provided a traditional argumentation on the relevance of CMRFs in CVD. However, the role of traditional risk factors over and above age in the population is being questioned because their low discriminatory accuracy. The low discriminatory accuracy of traditional risk factors is not a major problem as far as the treatment of risk factors does not harm. For instance, physical activity, healthy diet, reducing stress and an equitable distribution of resources is always recommended but pharmacological treatment could be questioned. I suggest the authors include a critical perspective in the introduction, rather than the usual “mantra” on the role CMRFs in CVD in the population. See for instance: Merlo J, Mulinari S, Wemrell M, Subramanian SV, Hedblad B. The tyranny of the averages and the indiscriminate use of risk factors in public health: The case of coronary heart disease. SSM Popul Health. 2017;3:684-98. The above publication refers to many others that may help to understand the idea.

AC 1.1 The authors are thankful to for the very constructive suggestion provided by the reviewer. After reading the Reviewer’s comments it was apparent that we had inadvertently over-emphasised considerations of individual- and area-level factors in the introduction, which suggested the aim of the study was to evaluate the importance of area-level factors in health service planning and provisioning. This was not our intent or the aim of the study, which was simply to describe geographic variation in cardiometabolic risk factors (CMRFs) in the study area. In light of this, we have substantially re-written the introduction to better reflect the intent of the research. 

Specific changes

1.1.1 In the revised manuscript, the Introduction now includes a critical perspective on inequalities in the geographic distribution CMRFs, and identified their potential to generate hypotheses on the individual- and area-level correlates of this variation. The changes are highlighted in the revised manuscript (with tracked in changes). Page: 4-5, lines 1- 41

1.1.2 We are also grateful to the reviewer in providing a useful source reference, which indicates the limitations of the current study. We have cited this work in the Discussion section under the limitations. Page: 4, line 14

RC 1.2 The authors also indicate, “Individual level approaches in the past have demonstrated limited success evidenced by its still increasing rates (of CVD), (7–9) area-level community health approaches, in addition, are important”. This is not new, and community level intervention has been launched in many countries since many years. The authors should review this issue and provide some references (to reviews of the literature). 

AC 1.2 We agree with the reviewer, and have reviewed on this and reported with references on previous reports.

 Specific Change

Reported on previous studies and reviews. Page 5, line: 28

RC 1.2.1 However, if you like to quantify the relative “importance” of the areas versus individuals for intervention you should identify the geographical component of the total individual risk variance by applying a multilevel approach. See for instance: Merlo J, Asplund K, Lynch J, Rastam L, Dobson A. Population effects on individual systolic blood pressure: a multilevel analysis of the World Health Organization MONICA Project. Am J Epidemiol. 2004;159(12):1168-79.

AC 1.2.1 As per our previous response, the study was designed for exploratory/descriptive analyses of any significant variation in the geographic distribution of CMRFs, which we have stated more clearly in the introduction section of the revised manuscript as it was ambiguous in the original manuscript. 

Specific change

1.2.1 Stated the main intent of study (page 5, lines 37 - 41). 

1.2.2 Referenced the article under future research in Discussion. Page13, line 219

RC 1.3 In addition, a main argument of the authors is that visualizing the geographical distribution of CMRFs has the potential to become a powerful toll for policy makers and health care service planners when planning area-level health care service. Again, this is not new. The use of GIS in health care has a long and well-established tradition. The news could be that the authors have developed an innovative GIS tool in their Region and this is always worthy. I do congratulate the authors. However, the authors should situate their work within this long and well-established GIS tradition.

AC 1.3 We thank the reviewer for recognising the potential contribution of this work to health care service policy initiatives. As suggested, we have altered this paragraph to appropriately situate the work within the well-established GIS domain.

Specific change

1.3.1 Repositioned the current study in the literature, and included additional references to reflect the analytical possibilities and limitations of GIS. Page: 5, lines 21–28

RC 1.4 My central concern, however, is the suitability of the analytical approach in this study. I agree in that “Discerning the distribution pattern of CMRFs in a given area is fundamental to such areal-level targeted approaches”. However, I think this “discerning” is not appropriately done by the use of GIS and spatial analyses. Those approaches are exclusively based on the analysis of area variance but question is when we can consider the area variance as large or important. Statistical significance is not a good criterion. To answer the question you need to discern the share of the total individual variance that is at the area level. For this purpose, you need to perform multilevel models and calculate some measures like the variance partition coefficient (VPC), the area under the curve (AUC) for random effect and even a measure of heterogeneity like the median odds ratio (MOR). Spatial analyses are much better analytical approach than traditional ecological analysis but they are not free of the ecological problems (fallacies, MAUP) and as the ecological analyses, they focus on geographical variation only. The “clustering” in spatial analyses refers to areas not to individuals within areas.

AC 1.4 The authors thank the reviewer for their comments and agree regarding the limitations of the analytical methods used in this study. As mentioned in our previous responses, the intent of the paper was not to “discern” the relative importance of individual- and area-level factors, but rather to elucidate the geographic variation in CMRFs. We apologise for the confusion generated in the Introduction, which has been clarified in the revised manuscript.

We would also like to disclose that the study is part of the first author’s PhD program, which is using different analytical methodologies to contrast and compare these data. We acknowledge that the level of evidence drawn out at different stages of analysis will vary; however, the staged nature of the research program precludes publishing all levels of evidence in a single paper. The multilevel analyses suggested by the reviewer are the focus of the next stage of this work, and we are grateful for their advice.

In response to the reviewer’s comments here, we have made clear of the limitations of this study design in the Discussion section. Also, a future research paragraph is added in the discussion section, where we have included and cited the analytical methodologies proposed by the reviewer in our future research based on the current results. 

Specific changes

1.4.1 Addressed the limitations of study methodology in Discussion. Page 13, lines 183–185

1.4.2 Included a paragraph on future research directions in Discussion. Page 13, lines 191–199

RC 1.5 GIS and spatial analysis are fancy and popular because coloured choropleth maps are attractive and easy to understand, and–for decisions makers– the cryptic language and scientific halo of the spatial analyses strength the authority of the information. However, they are ecological analyses with many pitfalls as compared with multilevel analyses. The problem with spatial analysis and GIS is that they may provide misguiding information to decision makers…. That is, just the opposite of what the authors aimed to do…. 

AC 1.5 We agree with the reviewers comments on the potential pitfalls of GIS methods and have now addressed this in the introduction with regards to GIS approaches in public health research. We have also included in the future research directions of this study that “Future research is required to understand the reasons for the geographic variation reported in this paper” 

Specific changes

 1.5.1 Addressed the pitfalls of GIS methods (page 4, lines 24–26).

 1.5.2 Stated the need for further explorations of current findings in Discussion. Page 13, line 191-192

RC 1.6 I refer to other publication for and extended explanation of my critics: 

 Merlo J, Wagner P, Leckie G. A simple multilevel approach for analysing geographical inequalities in public health reports: The case of municipality differences in obesity. Health Place. 2019;58:102145.

Merlo J, Viciana-Fernandez FJ, Ramiro-Farinas D, Research Group of Longitudinal Database of Andalusian P. Bringing the individual back to small-area variation studies: a multilevel analysis of all-cause mortality in Andalusia, Spain. Soc Sci Med. 2012;75(8):1477-87.

AC 1.6 The authors accept the very helpful references to analytical examples provided. We have included these references into our future research section.

Specific change

1.6.1 Included citations of suggested studies in future research in Discussion. Page 13, line 219

RC 1.7 The authors have a database that seems suitable for multilevel analyses, so my advice is to reanalyse the database with this methodology. You can see examples on how to perform the multilevel analyses together with codes and spreadsheets for calculations in previous publications, for instance: Merlo J, Wagner P, Ghith N, Leckie G. An Original Stepwise Multilevel Logistic Regression Analysis of Discriminatory Accuracy: The Case of Neighbourhoods and Health. PLoS One. 2016;11(4):e0153778.

 I also recommended the (free) information provided at the Centre for Multilevel Modelling, Bristol University http://www.bristol.ac.uk/cmm/ You can use the shrunken residuals from the multilevel analysis to create maps but you need to interpret the discriminatory accuracy of those maps by using the VPC and the AUC for instance.

AC 1.7 We agree with the reviewer on the suitability of our database for the suggested analyses, and we are very thankful for the detailed advice provided. As indicated in the previous responses, we intend to undertake these analyses in upcoming stages of our research. The intent of the current was to undertake a descriptive/explorative analysis, which we have now balanced with clear statements on its methodological limitations.

Specific change

1.7.1 Included citations of suggested analyses in future research section (Page 13, line 219).

RC 1.8 In summary, the authors have had a very worthy initiative by creating a record linkage database in their Region, The GIS and spatial analyses seems well performed but my concern is very fundamental. That is, the spatial GIS analyses are not suitable for your research question and, therefore, the conclusions you provide have not support. Your results and interpretation may even give misleading information to decision makers, as we do not know the discriminatory accuracy of the maps. To consider targeted area-level preventive interventions and regional health care service planning you need to perform a multilevel analysis as suggested above and considered the idea of proportionate universalism. Your database seems suitable and I recommend you read the analytical framework proposed above and reanalyse the data accordingly.

AC 1.8 We appreciate the recognition that the ‘GIS and spatial analyses seems well performed’, and the initiative taken in this study. The reviewer’s comments were very helpful to us in clarifying the analytical expectations which we had unintentionally created in the Introduction around the ‘quantification of variance’. The authors are grateful to the reviewer for identifying this ambiguity.

We acknowledge the descriptive approach of our study, and therefore its limitations with regards to ‘discriminatory accuracy’. We have now clearly framed the study within an exploratory/hypothesis generating domain of research, and moderate our claims to direct use in policy making in the revised Introduction and Discussion sections on limitations and future research. The current study will form the basis for further detailed multilevel analyses in the study region.

2. Response to reviewer # 2 comments: 

RC 2.1 This manuscript aims to assess spatial clustering patterns of CMRFs in the Illawarra Shoalhaven region of NSW Australia. The methods selected by the authors to do so are well-established ones and the resulting distribution patterns are insightful towards targeted local interventions.

AC 2.1 The authors are thankful to the reviewer for critically evaluating this work, their assessment of the appropriateness of the methods used and potential value of the reported findings for local interventions.

RC 2.2 The following may need to be clarified before the manuscript potentially moves forward to publication:

Data source: While the authors note the institutions which own the data & manage access, the link provided seems to be broken (https://www.ihmri.org.au/research-projects/simlr-cohortstudy/) and thus one would be unable to verify pertinent information.

AC 2.2 We thank the reviewer for identifying the broken link, which appears to have been updated since we submitted the manuscript. We have updated this in the revised manuscript, as follows:

Specific change

Made sure that the link is given/written right the Revised manuscript. The correct link is: https://www.ihmri.org.au/research-projects/simlr-cohort-study/ (page 14, line 243).

RC 2.3 The following may need to be clarified before the manuscript potentially moves forward to publication:

Table 2: Summary statistics of ACR & eGFR may need to be re-evaluated; specifically SD value of ACR and 3rd Qu & Max values of eGFR.

AC 2.3 Thank you raising these potential data errors, which are addressed below and in the manuscript where necessary.

Specific change

We have carefully reviewed the summary statistics values or ACR and eGFR in Table 2:

a) The SD value of the ACR is correct (SD = 40.3); however, there is a typographical error in the Max value of ACR in the original manuscript (Max = 91.5), which we have corrected in the Revised Manuscript (Max = 1291.50). The wider spread of ACR values in our data is the reason for the SD magnitude, which is noew correctly reflected by the updated Max value. Table 2, ACR Max value (corrected).

b) This was an oversight when we were preparing Table 2. The eGFR is truncated in the SIMLR Study database at >90 and assigned a value of 91 to indicate “normal” function. We have recalculated the mean, SD and median using grouped frequency data, and removed Q1 and Q3 for eGFR values in Table 2. Thank you for identifying this oversight. Table 2, eGFR summary values (corrected).

3. Response to reviewer # 3 comments:

RC 3.1 This paper uses pathology data to present the geographic distributions of a number of cardiovascular disease risk factors. There are problems with the underlying analysis and description of the analysis. These problems are severe enough that they must be addressed prior to a more detailed review:

AC 3.1 The authors are thankful to the reviewer 3 for their critical evaluation of this work and helpful suggestions for improvement.

RC 3.1.1 The denominator used is the paper is unclear and is described differently at multiple points in the paper. 

a) The title indicates that this analysis uses 256,385 adult individuals.

b) This value again appears in the results (line 193).

c) However, Table 2 indicates that there are 256,385 tests, not individuals.

d) However, the methods (line 167) indicate that there are 256,385 tests, not individuals. This is almost certainly the case.

e) The authors report a population of 285,385 adults. The data source is a private pathology service, not a population-level registry or health record. These numbers would then indicate that almost 90% of individuals living in the study area had at least one lab test from this pathology company.

f) Another example of this confusion between number of tests and number of adults is the GFR test. 244,166 tests are reported from 256,525 adults. These values indicate that 85% of the adult population received a GFR test, which almost certainly cannot be correct. 

g) However, if the denominator of the described proportions is the number of tests, then these values cannot be interpreted as the authors have.

AC 3.1.1 Thank you for raising these issues. While we have been consistent throughout the manuscript that the sample giving rise to the tests included in this study is 256,525, we realise we have used denominators inconsistently, which and this has given rise to the confusion. To clarify, the study base for this analysis comprises 256,525 unique individuals who have contributed a total of 1,132,016 test results. We have removed the percentages in Table 2 that were mixing numerator (tests) and denominator (persons) sources, and have included a textual narrative at the beginning of the result sections indicating: (1) the total number of tests (1,132,016); (2) the total number of individuals contributing these tests (256,525); and (3) the number and percentage of individuals with a result for each of the tests included in Table 2. We sincerely apologise for the confusion our inexactness has caused. 

Specific actions

3.1.1.1 Add textual narrative to results section (page 9, lines 136–141)

3.1.1.2 Removed percentages from Table 2 (page 9, Table 2: column 2) 

3.1.1.4 Updated footnote for Table 3 (page 10, line 150).

RC 3.2 The authors refer to the calculated measures as “risk”. In epidemiology, this term has a specific meaning that implies using person-time as a denominator. Had the authors used population as a denominator, they would have been calculating prevalence, not risk. However, if (as seems to be true) the authors used the number of tests as the denominator, then these values are simply the proportion of tests meeting a specific criteria.

AC 3.2 Thank you for identifying our inexact use of risk, which was not intended in the epidemiological sense but with regards to the “CMRF risk classification” defined in Table 1 (page 7, line 105). We have gone through the manuscript and clarified the use of ‘risk’ in reference to CMRF ‘higher risk’ classification.

Specific action

3.2.1 The usage of word ‘risk’ has been changed to define ‘higher risk’ classification throughout the manuscript. Page 7, line 105; Page 8, line 111, 113; Page 9 line 144,145,146; Page 19 line 149; Page 13 line 199

RC 3.3 Since these tests were performed on an unhealthy population (as evidenced by 95% having had a kidney function test), these results cannot be generalized to the total population. The results should then be interpreted in light of this.

AC 3.3 We agree with the reviewer 3 that these results may not be generalisable, and have noted this possibility as study limitation in the Discussion.

Specific change

3.3.1 Stated in the limitations of the study. Page 13, lines 186 -188

RC 3.4 Overall, the writing in this article is very poor. There are multiple instances of sentences and phrases that are convoluted and have an unclear meaning. I would recommend that an editor and an experienced epidemiologist review this manuscript before publication to clarify the manuscript and to ensure that correct epidemiologic terms are used.

3.4.1 One example of this is the last sentence of the introduction. I’m unsure what the authors are trying to convey in that sentence and why that sentence is in the paper. 

3.4.2 There are multiple, similar examples throughout the paper.

a) e.g. “frequency and proportion of risks” (line 166), 

b) “discrete test proportions” (line 194), etc.).

AC 3.4 We thank the reviewer for bringing this to our attention, and have sought to address these in the revised manuscript.

 Specific changes

3.4.1 Changed the last sentence in the Introduction section (page 5, line 39-41).

3.4.2 We have carefully gone through the revised manuscript and sought to identify and correct similar instances of usages indicated by the reviewer that are convoluted or have an unclear meaning (changes highlighted throughout the revised manuscript).

RC 3.5 Finally, I question if it is reasonable to assume that individuals in the lab data are equally distributed throughout the study area. Spatial clustering of some results could simply be the result of some areas being over-represented in the data. That is, some locations with a high proportion of positive tests could reflect greater access to the pathology lab in that location.

AC 3.5 We thank the reviewer for raising this concern. For commercial in-confidence reasons we are unable to indicate in the manuscript that this laboratory provides >90 of the private pathology services in the study area, and has an organisational commitment to equity of access that is realised through an extensive and diverse collection centre network that provides population coverage for the study area. The laboratory also provides Medicare bulk-billed services where this is indicated on the request form. While we agree with the reviewer that individuals in the laboratory data need not be distributed equally in the study area, this more likely represents the geographic distribution of the population across the study area rather than a testing bias due to greater access. However, we cannot exclude the possibility a that higher proportion of positive tests in an area could be due to greater access to pathology labs. This is beyond the scope of the current study but we thank the reviewer for raising this possibility and have included it as a potential limitation in the Discussion.

Specific changes

3.5 Added this as a limitation in Discussion (Page 13, line 188-190)

---

## [Decision Letter · Decision Letter 1]

17 Sep 2019

Geographic variation in cardiometabolic risk distribution: a cross sectional study of 256,525 adult residents in the Illawarra-Shoalhaven region of the NSW, Australia.

PONE-D-19-14817R1

Dear Dr. Toms,

We are pleased to inform you that your manuscript has been judged scientifically suitable for publication and will be formally accepted for publication once it complies with all outstanding technical requirements.

With kind regards,

Wenhao Yu, Ph.D.

Academic Editor

PLOS ONE

Additional Editor Comments (optional):

Reviewers' comments:

Reviewer's Responses to Questions

**Comments to the Author**

1. If the authors have adequately addressed your comments raised in a previous round of review and you feel that this manuscript is now acceptable for publication, you may indicate that here to bypass the “Comments to the Author” section, enter your conflict of interest statement in the “Confidential to Editor” section, and submit your "Accept" recommendation.

Reviewer #1: All comments have been addressed

Reviewer #2: All comments have been addressed

Reviewer #3: All comments have been addressed

2. Is the manuscript technically sound, and do the data support the conclusions?

Reviewer #1: Yes

Reviewer #2: Yes

Reviewer #3: (No Response)

3. Has the statistical analysis been performed appropriately and rigorously? 

Reviewer #1: Yes

Reviewer #2: Yes

Reviewer #3: (No Response)

4. Have the authors made all data underlying the findings in their manuscript fully available?

Reviewer #1: (No Response)

Reviewer #2: Yes

Reviewer #3: (No Response)

5. Is the manuscript presented in an intelligible fashion and written in standard English?

Reviewer #1: Yes

Reviewer #2: Yes

Reviewer #3: (No Response)

6. Review Comments to the Author

Reviewer #1: This is a carefully performed study using data with limitations. However, the author performs an excellent work including a critical exposure of the weaknesses of the study. The aims and conclusions are prudent and balanced and the authors suggest future lines of investigation. In this case the information need be validate, the coverage of the register identified, and the analyses should be done by multilevel regression analysis. However, the present paper as it is now provides worthy information.

Reviewer #2: (No Response)

Reviewer #3: (No Response)

7. PLOS authors have the option to publish the peer review history of their article (what does this mean?). If published, this will include your full peer review and any attached files.

Reviewer #1: No

Reviewer #2: No

Reviewer #3: Yes: Adam S. Vaughan

---

## [Editor Report · Acceptance letter]

20 Sep 2019

PONE-D-19-14817R1 

Geographic variation in cardiometabolic risk distribution: a cross sectional study of 256,525 adult residents in the Illawarra-Shoalhaven region of the NSW, Australia. 

Dear Dr. Toms:

I am pleased to inform you that your manuscript has been deemed suitable for publication in PLOS ONE. Congratulations! Your manuscript is now with our production department. 

With kind regards,

on behalf of

Dr. Wenhao Yu 

Academic Editor

PLOS ONE